# Do Patient-Reported Outcome Measures (PROMs) Used Within Radiotherapy Clinical Trials Reflect the Impact of Treatment?

**DOI:** 10.3390/cancers16223832

**Published:** 2024-11-14

**Authors:** Danielle Fairweather, Rachel M. Taylor, Laura Allington, Nazima Haji, Naomi Fersht, Yen-Ching Chang, Rita Simões

**Affiliations:** 1Cancer Division, University College London Hospitals NHS Foundation Trust, London NW1 2BU, UK; l.allington@nhs.net (L.A.); rita.simoes@nhs.net (R.S.); 2Centre for Nurse, Midwife and AHP Research, University College London Hospitals NHS Foundation Trust, London NW1 2PG, UK; rtaylor13@nhs.net; 3Department of Targeted Intervention, University College London, London WC1E 6BT, UK

**Keywords:** patient-reported outcome measures, side-effects, radiotherapy, proton beam therapy, clinical outcomes, clinical trials

## Abstract

Patient-reported outcome measures (PROMs) are used in radiotherapy to evaluate the side-effects that patients may experience and how treatment can affect a patient’s quality of life. They consist of validated questionnaires that are filled in by patients at several timepoints during or after treatment. Our study assessed if PROMs used in radiotherapy trials are measuring the impact of treatment on patients. We found that 45 out of 51 PROMs (88%) are missing common side-effects that patients who have radiotherapy experience, meaning that important differences in how patients experience these treatments might be missed. Future work in PROMs is needed to develop accurate tools that measure how radiotherapy impacts patients.

## 1. Introduction

Radiotherapy is widely used in cancer treatment, with 40% of patients receiving it as part of their overall care. However, many of these patients face both immediate and long-term physical and psychological side-effects, which can significantly affect their health-related quality of life (HR-QoL) [1]. Proton beam therapy (PBT) offers potential for lowering the risk of long-term side-effects, though additional phase III trials are needed to confirm its effectiveness [2,3]. Robust ways of assessing and reporting patient outcomes are essential to demonstrate treatment effectiveness, patient satisfaction, and improvements to quality of life [4,5]. Traditional clinician-reported outcomes have been demonstrated to under-report the treatment burden, and therefore, implementing patient-centred data reporting is essential for determining a truer sense of treatment side-effects [6].

PROMs are ‘standardised, validated questionnaires that are completed by patients in order to measure their perceptions of their own functional status and wellbeing’ [1]. Within the last few years, a great deal of emphasis has been placed on the use of patient-reported outcome measures (PROMs) and the importance of their role within clinical trials and research [7]. They are designed to capture concepts related to the health experiences of an individual and can be used to monitor the disease impact and treatment response, as well as aid patient–clinician communication and decision making [8,9]. PROMs are also an important outcome which can be utilised within clinical trials to capture and quantify the benefits or risks of treatments in research studies [10].

PROMs are highly utilised within chemotherapy and drug trials, as recommended by the US Food and Drug Administration (FDA) [9]. Despite the increasing efforts to incorporate PROMs into radiotherapy clinical trials, they remain somewhat under-adopted as primary or secondary outcomes [6]. This could be due to the fact the majority of PROMs are generic, focusing on overall HR-QoL, and therefore are not specific to the impacts of radiotherapy or PBT [1,11]. In a previous review completed by Fairweather et al., PROMs were explored for use within routine clinical practise within radiotherapy and PBT [12]. In this article, we instead aim to investigate the appropriateness of PROMs and their use within radiotherapy clinical trials, comparing PROMs with toxicity-based trial endpoints. This comparison of PROMs and treatment toxicities aims to identify which PROMs are appropriate for use within radiotherapy and PBT clinical trials, aiding clinicians and researchers in choosing the most appropriate PROM for their study.

## 2. Materials and Methods

In this paper, we focused on the six most common tumour sites treated with radiotherapy and PBT in the United Kingdom (UK): the breast, brain, head and neck (H&N), lung, prostate, and sarcoma [13,14]. The medical research platform OVID was used to search the bibliographic databases MEDLINE, EMBASE, and EMCARE. Search terms were identified using the PICO (Patient, Intervention, Comparator, Outcome) framework, and searches were limited to publications from the 1 January 2008 to 1 June 2024. Published articles were eligible for inclusion if they were written in English and if the participants involved were human, aged 16 and above, and treated with external beam radiotherapy or PBT for the six previously described tumour sites. The articles included described the use of PROMs as an outcome measure within clinical research. All observational and interventional studies were eligible.

The identified PROMs were extracted into Microsoft Excel (Version 16.90.2) and categorised into groups based on the patient cohort or treatment site. The quality of the PROMs in previous research was appraised using the COSMIN (COnsensus-based Standards for the selection of health Measurement INstruments) reporting guidelines [12].

Site-specific side-effects were extracted from the consent forms and guidelines published by the UK Royal College of Radiologists (RCR) (www.rcr.ac.uk, accessed on 8 July 2024) [15]. For each of the six treatment sites, the corresponding RCR National Radiotherapy consent form was used to extract the expected (50–100%), common (10–50%), less common (>10%), acute, and late side-effects. The RCR’s supporting development document, which accompanies the consent forms, was used to confirm that the reasoning and use of plain language were appropriate. The articles, PROMs, and site-specific side-effects were independently screened, checked for eligibility, and extracted by two reviewers (DF, RS), with any discrepancies discussed with a third reviewer (RT) in the case of disagreement. The items within each PROM were scored with a Yes or No, for presence or absence, against the site-specific side-effects recommended for the treatment site [15] and recorded within a Microsoft Excel spreadsheet. The results are presented descriptively.

## 3. Results

In total, 51 different PROMs were identified (Appendix A). The PROMs were categorised by treatment location, with ten PROMs being used for patients with breast cancer, seven in the brain, sixteen in H&N, four in the lungs, seven for sarcoma, and seventeen in the prostate (Table 1, Table 2, Table 3, Table 4, Table 5 and Table 6). There were four PROMs that were utilised across more than one treatment site. The European Organisation for Research and Treatment of Cancer (EORTC) Quality of Life Questionnaire (QLQ)-C30 was used across all six tumour types. The Hospital Anxiety and Depression Scale (HADS) was utilised for breast, brain, and H&N cancers. The EuroQol Group EQ-5D-5L questionnaire, which is classified as a generic HR-QoL measure, was used for breast, brain, lung, and prostate cancers. The Patient-Reported Outcomes version of the Common Terminology Criteria for Adverse Events (PRO-CTCAE) was used with both H&N and lung cancer patients.

### 3.1. Breast Cancer

Table 1 highlights the side-effects listed in the RCR guidelines for the treatment of the breast [15] compared to the items found in the ten PROMs found in the literature. Fatigue and hair loss in the treatment area are an expected side-effect of radiotherapy [15]. As shown in Table 1, five PROMs included a question on tiredness. There was only one PROM that captured hair loss: the EORTC-QLQ-BR23. Skin soreness, itching, blistering, and colour changes in the treatment area are also common side-effects of radiotherapy [15]. There were two PROMs that captured these toxicities: the EORTC-QLQ-BR23, a breast-specific questionnaire, and an unvalidated breast-specific questionnaire developed as part of the IMPORT-LOW clinical trial.

Discomfort in the treatment area, breast swelling, and changes in breast texture are all reported as less common side-effects of breast radiotherapy [15]. As highlighted in Table 1, pain and discomfort were captured within four PROMs. Swelling was captured within both the EORTC-QLQ-BR23 and the IMPORT-LOW questions. Change in breast texture was only captured by the IMPORT-LOW questions. Other rare side-effects of breast radiotherapy that were included were a sore throat and pneumonitis, which can lead to a cough or breathlessness [15]. The only one of these which was captured was shortness of breath in the EORTC-QLQ-C30.

**Table 1 cancers-16-03832-t001:** A comparison of breast radiotherapy side-effects and patient-reported outcome measures.

	PROM									
Site-Specific Side-Effects	EORTC-QLQ-C30	EORTC- QLQ-BR23	HADS	Body Image Scale	IMPORT LOW Trial	EQ-5D-5L	Revised Piper Fatigue Scale	C-QOL	Cancer Fatigue Scale	SF-8
Tiredness	Y						Y	Y	Y	Y
Hair loss		Y								
Skin reaction		Y			Y					
Skin itching		Y								
Pain or discomfort		Y				Y		Y		Y
Swelling or lymphoedema		Y			Y					
Change in breast texture					Y					
Change in skin colour					Y					
Change in breast appearance		Y		Y	Y					
Cough										
Shortness of breath	Y									
Sore throat or dysphagia										
Shoulder stiffness		Y								
Tingling or numbness in arm										

BR23—Breast cancer module; C30—General cancer module; C-QOL—Quality of Life scale for Korean patients with cancer; EORTC—European Organisation for Research and Treatment of Cancer; EQ—EuroQoL group; HADS—Hospital Anxiety and Depression Scale; QLQ—Quality of Life Questionnaire; SF—Short form.

### 3.2. Brain Tumours

The common side-effects in brain radiotherapy or PBT were compared to the seven identified PROMs (Table 2). Fatigue and pain (such as headaches) are commonly experienced by patients who receive radiotherapy or PBT to the brain [15,16]. Tiredness/a lack of energy/fatigue was captured within five of the PROMs, and pain was captured within three of those. The EORTC-QLQ-BN20, a brain-specific EORTC module, captures skin itching and hair loss. None of the seven PROMs captured skin reactions such as changes in skin colour, skin soreness, dry desquamation, or moist desquamation [17].

Nausea, vomiting, and a loss of appetite are commonly experienced by patients receiving radiotherapy or PBT to the brain [15]. There were two PROMs that captured nausea: the EORTC-QLQ-C30 and Functional Assessment of Cancer Therapy (FACT)-Br. Vomiting and loss of appetite were captured by one PROM, the EORTC-QLQ-C30. Other less common side-effects of radiotherapy or PBT to the brain such as cognitive impairment and reduced motor function/weakness were captured by FACT-Br, the Chalder Fatigue Questionnaire, and EORTC-QLQ-BN20 [15,16]. FACT-Br was the only PROM to capture changes in hearing. The HADS and GAD-7 questionnaires are a measure of anxiety and/or depression and therefore did not capture any site-specific side-effects of brain radiotherapy.

**Table 2 cancers-16-03832-t002:** A comparison of brain radiotherapy and proton beam therapy side-effects and patient-reported outcome measures.

	PROM						
Site-Specific Side-Effects	EORTC-QLQ-C30	EORTC-QLQ-BN20	HADS	FACT-BR	Chalder Fatigue Questionnaire	GAD-7	PHQ-9
Tiredness	Y	Y		Y	Y		Y
Hair loss		Y					
Skin reaction							
Skin itching		Y					
Pain or discomfort	Y	Y		Y			
Nausea	Y			Y			
Vomiting	Y						
Loss of appetite	Y						Y
Weakness or reduced motor function		Y		Y	Y		
Changes in vision		Y		Y			
Changes in hearing				Y			
Cognitive impairment		Y		Y	Y		Y
Seizures		Y		Y			

BN20—Brain tumour module; C30—General cancer module; EORTC—European Organisation for Research and Treatment of Cancer; EQ—EuroQoL group; GAD-7—Generalised Anxiety Disorder Questionnaire; HADS—Hospital Anxiety and Depression Scale; FACT—Functional Assessment of Cancer Therapy; PHQ-9—Patient Health Questionnaire; QLQ—Quality of Life Questionnaire.

### 3.3. Head and Neck Cancer

The common side-effects of H&N radiotherapy or PBT are compared to the items assessed in 16 PROMs in Table 3. The concept of fatigue was captured within seven of the PROMs. However, there was variation between different wordings when describing the concept of fatigue: FACT-HN uses ‘lack of energy’, the EORTC-QLQ-C30 uses ‘tiredness’, and the PRO-CTCAE questionnaire uses ‘lack of energy, fatigue, and tiredness’ in one question. The Head and Neck Radiotherapy Questionnaire (HNRQ) asks two separate questions, one on fatigue and tiredness and another on ‘lack of energy’. Although these terms are categorised together within Table 3, it is important to note that these concepts may be perceived differently [18,19].

Only three of the PROMs captured radiation-induced skin reactions: PRO-CTCAE, the MD Anderson Symptom Inventory for H&N cancer (MDASI-HN), and HNRQ. Skin itching was captured by only two of those PROMs, and hair loss was only captured by one, the PRO-CTCAE (Table 3). Pain is another common side-effect and can be particularly severe within the oral cavity [15], and this was included in ten PROMs. Dysphagia, another common symptom, was found within 11 of the PROMs. Thick saliva, dry mouth, cough, and voice changes, which are site-specific side-effects for H&N cancer treatment, were all incorporated in the EORTC-QLQ-HN35 and MDASI-HN. Altered taste or smell was captured within seven PROMs. Nausea and vomiting were included within the EORCT-QLQ-C30, MDASI-HN, PRO-CTCAE, and HNRQ. Dental problems were captured within three PROMs, but trismus was only measured in the EORTC-QLQ-HN35 (Table 3).

**Table 3 cancers-16-03832-t003:** A comparison of head and neck radiotherapy and proton beam therapy side-effects and patient-reported outcome measures.

	PROM															
Site-Specific Side-Effects	EORTC-QLQ-C30	EORTC-QLQ-HN35	EORTC-QLQ-OH15	MDASI-HN	MDADI	UW-QOL	HADS	FACT-HN	SWAL-QOL	S-SECEL	SQLI	HNRQ	PRO-CTCAE	XeQoLS	OHIP-14	PROMS Scale
Tiredness	Y			Y		Y		Y			Y	Y	Y			
Hair loss													Y			
Skin reaction				Y								Y	Y			
Skin itching												Y	Y			
Pain or discomfort	Y	Y	Y	Y		Y		Y				Y	Y		Y	Y
Nausea	Y			Y				Y				Y	Y			
Vomiting	Y			Y								Y	Y			
Loss of appetite	Y			Y					Y			Y	Y			
Dry mouth		Y	Y	Y		Y		Y				Y	Y	Y		
Cough		Y		Y	Y				Y				Y			
Thick saliva		Y	Y	Y		Y			Y			Y				
Voice changes		Y		Y				Y		Y		Y	Y			
Dysphagia		Y	Y	Y	Y	Y		Y	Y			Y	Y		Y	Y
Dental problems		Y	Y	Y												
Change in skin colour				Y									Y			
Trismus		Y														
Altered taste or smell		Y	Y	Y		Y						Y			Y	Y
Weight loss		Y			Y											
Swelling		Y											Y			

C30—General cancer module; EORTC—European Organisation for Research and Treatment of Cancer; EQ—EuroQoL group; FACT—Functional Assessment of Cancer Therapy; HADS—Hospital Anxiety and Depression Scale; HN35—Head and Neck cancer module; HNRQ—Head and Neck Radiotherapy Questionnaire; MDADI—MD Anderson Dysphagia Inventory; MDASI-HN—MD Anderson Symptom Inventory for Head and Neck Cancer; OH15—Oral Health module; OHIP-14—Oral Health Impact Profile; PRO-CTCAE—Patient-Reported Outcomes version of the Common Terminology Criteria for Adverse Events; PROMS scale—Patient-Reported Oral Mucositis Symptom Scale; QLQ—Quality of Life Questionnaire; SQLI—Spitzer Quality of Life Index; S-SECEL—Swedish Self-Evaluation of Communication Experiences after Laryngeal problems; SWAL-QOL—Swallowing Quality of Life Questionnaire; UW-QOL—University of Washington Head and Neck Quality of Life Questionnaire; XeQoLS— Xerostomia Quality of Life Scale.

### 3.4. Lung Cancer

Four PROMs were identified as being used with patients receiving radiotherapy to the lung (Table 4). Pain or discomfort was included in all four PROMs. This was the only site-specific side-effect that was captured by EQ-5D-5L (Table 4). Tiredness, nausea, and vomiting were measured in the EORTC-QLQ-C30 and PRO-CTCAE. Only one of the PROMs, PRO-CTCAE, included items that captured radiation-induced skin reactions or skin itching. The EORTC-QLQ-LC13 lung module was the only PROM that captured hair loss as a symptom. The EORTC-QLQ-C30, EORTC-QLQ-LC13, and PRO-CTCAE captured shortness of breath, but only the -LC13 module and PRO-CTCAE went on to include coughing and dysphagia (Table 4).

**Table 4 cancers-16-03832-t004:** A comparison of lung side-effects and patient-reported outcome measures.

	PROM			
Site-Specific Side-Effects	EORTC-QLQ-C30	EQ-5D-5L	EORTC-QLQ-LC13	PRO-CTCAE
Tiredness	Y			Y
Hair loss			Y	
Skin reaction				Y
Skin itching				Y
Pain or discomfort	Y	Y	Y	Y
Nausea	Y			Y
Vomiting	Y			Y
Cough			Y	Y
Shortness of breath	Y		Y	Y
Dysphagia			Y	Y
Haemoptysis			Y	
Tingling or numbness in arm			Y	Y
Change in skin colour				Y

C30—General cancer module; EORTC—European Organisation for Research and Treatment of Cancer; EQ—EuroQoL group; LC13—Lung cancer module; PRO-CTCAE—Patient-Reported Outcomes version of the Common Terminology Criteria for Adverse Events; QLQ—Quality of Life Questionnaire.

### 3.5. Sarcoma

As shown in Table 5, seven PROMs were identified as being used for patients with sarcoma. The sarcoma category was further categorised into treatment site to best compare site-specific side-effects. Three were used within ‘pelvic sarcoma’, and four PROMs were used within ‘limb sarcoma’ and are therefore categorised as such.

Tiredness was captured in four PROMs (Table 5). Radiation-induced skin reactions and skin itching were not included in any PROM. EORTC-QLQ-CR29 is a colorectal specific EORTC module which was being used with patients with pelvic sarcoma as part of the PROSPER study, which aims to compare the effectiveness of carbon ion therapy, PBT, and surgical intervention [20]. It contains items that refer specifically to skin soreness around the anus or stoma bag. However, these questions do not reflect radiation-induced skin reactions in other areas, such as moist desquamation within skin folds.

Pain and discomfort were measured within all seven PROMs. Swelling and lymphoedema were not captured within any of the PROMs. Joint stiffness, weakness, or reduced motor function was captured in all the limb sarcoma PROMs (Table 5). Nausea, vomiting, and diarrhoea were only included in the EORTC-QLQ-C30. Increased urinary and bowel frequency, urgency or incontinence, and blood in urine or stool were only captured in the EORTC-QLQ-CR29.

**Table 5 cancers-16-03832-t005:** A comparison of sarcoma radiotherapy and proton beam therapy side-effects and patient-reported outcome measures.

		PROM					PROM			
	Site-Specific Side-Effects	EORTC-QLQ-C30	PROMIS-29	EORTC-QLQ-CR29		Site-Specific Side-Effects	MHQ	FAOS	TESS-UE	TESS-LE
PELVIC SARCOMA	Tiredness	Y	Y		LIMB SARCOMA	Tiredness			Y	Y
	Increased urinary frequency or urgency			Y		Hair loss				
	Increased bowel frequency or urgency			Y		Skin reaction				
	Dysuria					Skin itching				
	Pain or discomfort	Y	Y	Y		Pain or discomfort	Y	Y	Y	Y
	Looser stools	Y				Change in skin colour				
	Skin reaction					Change in skin texture				
	Skin itching					Joint stiffness	Y	Y	Y	Y
	Nausea	Y				Weakness or reduced motor function	Y	Y	Y	Y
	Vomiting	Y				Swelling or lymphoedema		Y		
	Urinary incontinence			Y		Tingling or numbness	Y			
	Faecal incontinence			Y		Loss of appetite				
	Blood in stool or urine			Y						
	Early menopause									
	Swelling or lymphoedema									
	Hair loss			Y						
	Vaginal stenosis			Y						
	Ability to achieve and maintain erections			Y						

C30—General cancer module; CR29—Colorectal module; EORTC—European Organisation for Research and Treatment of Cancer; FAOS—Feet and Ankle Outcome Score; LE—Lower extremity; PROMIS—Patient-Reported Outcomes Measurement Information System; QLQ—Quality of Life Questionnaire; MHQ—Michigan Hand Outcomes Questionnaire; TESS—Toronto Extremity Salvage Score; UE—Upper Extremity.

### 3.6. Prostate Cancer

The common side-effects for radiotherapy to the prostate were compared to the items within 17 PROMs (Table 6). Tiredness was captured in eight PROMs. Increased urinary frequency or urgency was included in ten, increased bowel frequency or urgency were included in seven, and loose stools were captured in six PROMs.

The ability to achieve and maintaining erections, changes in ejaculation, and loss of orgasm are also known side-effects of prostate radiotherapy. These were only measured in two PROMs in their entirety: the EORTC-QLQ-PR19 and its updated version the EORTC-QLQ-PR25. The Expanded Prostate Cancer Index Composite (EPIC) questionnaire, its shortened counterpart EPIC-26, the International Index of Erectile Dysfunction (IIEF-5), and the UCLA Prostate Cancer Index (UCLA-PCI) questionnaires captured the ability to achieve and maintaining erections and loss of orgasm, but not changes in ejaculation.

Radiation-induced skin reactions, as well as pain or discomfort, are less common side-effects of prostate radiotherapy, as well as tenesmus and blood in urine or stool [15]. Tenesmus was measured in one PROM—FACT-P. Skin reactions were only captured in the EORTC-QLQ-PR19.

**Table 6 cancers-16-03832-t006:** A comparison of prostate radiotherapy side-effects and patient-reported outcome measures.

	PROM																
Site-Specific Side-Effects	EORTC-QLQ-C30	EORTC-QLQ-PR19	EORTC-QLQ-PR25	EPIC	EPIC-26	SF-12	SF-36	FACT-P	Hoffman et al. [21]	UCLA PCI	SWOG-QoL	EQ-5D-5L	HFR-DIS	IPSS	IIEF-5	ICIQ-UI SF	PROMIS-Fatigue
Tiredness	Y			Y	Y	Y	Y	Y			Y						Y
Increased urinary frequency or urgency		Y	Y	Y	Y			Y	Y	Y	Y			Y		Y	
Increased bowel frequency or urgency		Y	Y	Y	Y				Y	Y	Y						
Dysuria		Y	Y	Y	Y			Y	Y					Y			
Pain or discomfort	Y		Y	Y	Y	Y	Y	Y		Y	Y	Y					
Looser stools	Y	Y		Y					Y	Y	Y						
Skin reaction		Y															
Tenesmus								Y									
Blood in stool or urine		Y	Y	Y	Y				Y								
Urinary incontinence		Y	Y	Y	Y			Y	Y	Y	Y					Y	
Faecal incontinence		Y	Y	Y	Y				Y								
Change in ejaculate		Y	Y														
Ability to achieve and maintain erections		Y	Y	Y	Y			Y	Y	Y					Y		
Loss of orgasm		Y	Y	Y	Y					Y					Y		
Swelling or lymphoedema		Y	Y														
Hair loss				Y													

C30—General cancer module; EORTC—European Organisation for Research and Treatment of Cancer; EPIC—Expanded Prostate Cancer Index Composite; EQ—EuroQoL group; FACT—Functional Assessment of Cancer Therapy; HFR-DIS—Hot-Flash-Related Daily Interference Scale; ICIQ-UI SF—International Consultation on Incontinence Questionnaire—Urinary Incontinence Short Form; IIEF-5—International Index of Erectile Function; IPSS—International Prostate Symptom Scale; PR-19—Prostate module; PR-25—updated Prostate module; PROMIS—Patient-Reported Outcomes Measurement Information System; QLQ—Quality of Life Questionnaire; SF—Short-form survey; SWOG-QoL—South Western Oncology Group Quality of Life Questionnaire; UCLA-PCI—University of California, Los Angeles Prostate Cancer Index.

## 4. Discussion

To our knowledge, this is the first review where the appropriateness of PROMs for detecting radiotherapy side-effects as an endpoint within clinical trials has been investigated. All the PROMs found in the literature search are widely used in clinical trials and research. However, none of them appear to capture all the symptoms listed in the RCR guidelines, independently of the tumour site. Using non-specific PROMs carries the risk of not capturing important differences between treatment methods [12,22]. Researchers and healthcare professionals are advised to carefully select the PROMs that report the appropriate outcomes that are linked to trial endpoints [22,23].

The use of PROMs within research is recommended by the FDA, as they allow us to understand the impact of a treatment from the patients’ perspective [9]. In guidance developed by the FDA, it is highlighted that when choosing a PROM within clinical trials, there should be evidence that shows that it can measure the concept it is intended to measure, i.e., that it is valid [9]. This review highlights that there are a wide variety of PROMs being utilised within radiotherapy and PBT research, but most lack specificity to radiotherapy and PBT side-effects. Consequently, radiotherapy and PBT healthcare professionals are not able to measure common impacts of a treatment. PROMs should be sensitive to change over time, but this is only possible if the items within the PROMs are able to capture the changes that occur within a specific cohort of patients or situation [24].

Many generic HR-QoL measures were developed some time ago. For example, the EORTC-QLQ-C30, which was utilised across a multitude of treatment sites, was designed in 1993 [25]. Similarly, the EQ-5D was developed in the 1980s [26], and SF-36 in 1992, which was then evolved into the shortened version, SF-12 in 1996 [27,28]. The landscape of cancer treatment in the modern day has significantly changed since the development of these historic measures. The technological advances in radiotherapy and PBT have also since resulted in significant improvements in symptoms, HR-QoL, and survivorship [1,11]. As described by Faithful et al., it is not uncommon for the questionnaires to evolve and undergo updates over time (none of the measures above have been updated), but this review highlights that special focus should be given to include the modern-day impacts of radiotherapy and PBT going forward, considering that they are treatments received by 40% of cancer patients [1].

A key side-effect that is missing from the reviewed PROMs was related to radiation-induced skin reactions. Symptoms such as changes in skin colour, itching, dry and moist desquamation, and other visible changes at the treatment site affect up to 95% of radiotherapy patients [17]. However, only 12% of the reviewed PROMs included items addressing these skin reactions. In a recent content validity study of the EORTC-QLQ-C30, cancer patients identified skin issues as a concern during interviews, yet this symptom is not captured in the questionnaire [29]. The EORTC-QLQ-C30 is commonly used within radiotherapy and PBT clinical trials despite this important omission. The EORTC recently published the development of a ‘Write in three Symptoms/Problems’ instrument that can be used to capture additional symptoms that are not assessed within the currently available EORTC questionnaires [30]. Within the acceptability study, skin problems were the most commonly reported symptom that patients undergoing active treatment would input into the instrument [30]. However, the utility of this within a standardised clinical trial protocol and analysis plan has yet to be defined, especially as there may be variation in the way patients interpret the wording of side-effects.

The concept of fatigue is another universally experienced side-effect of radiotherapy and PBT, regardless of treatment site [15]. Fatigue, and other variations of the concept, are captured in 50% of the PROMs in this review. However, only ten of the PROMs found within this review utilise the actual word fatigue. The EORTC-QLQ-C30, for example, uses the word ‘tired’, whereas the SF-12 asks about a ‘lack of energy’ [25,28]. Some of the PROMs, such as HNRQ, ask separate questions about fatigue and a ‘lack of energy’. As Richardson et al. highlighted, these variations of describing fatigue can be perceived as a different concept entirely and may not capture the true nature of fatigue. Asking about the impact of fatigue, rather than the characteristic, may improve our ability to capture the concept [18,19].

When selecting PROMs, it is important to understand the theoretical basis for the measures being used, the outcomes that need to be captured, and which PROMs capture them [22,31]. Ask the question, is there evidence that the PROM being used reliably measures the concept within the patient population enrolled within the study? Is the PROM specific to the concept you wish to study and sensitive enough to reflect the patient’s experience [9]? Items that are ‘Irrelevant’ can reduce the content validity and interpretability of PROMs [32]. Patients may feel frustrated by questions that seem unrelated to their experiences, which can lead to biassed responses or reduced response rates [10]. Likewise, combining PROMs can result in multiple overlapping questions, which can leave patients feeling frustrated, and although completing multiple questionnaires can allow for an exploration of global and specific effects, it is an added time burden that could affect acceptability [1,10].

The Proton Clinical Outcomes Unit has previously emphasised the value of incorporating PROMs into PBT clinical outcome data collection [3]. PBT is anticipated to offer many patients an improved HR-QoL compared to existing radiotherapy techniques, yet there is currently limited evidence to support this claim [2,3]. There are many proton vs. photon clinical trials that are currently ongoing and that utilise PROMs as a primary outcome [2], but as this review highlights, the PROMs currently being used within radiotherapy and PBT research may not be sensitive enough to detect differences. Until new or improved PROMs are available, great thought and caution should be taken when selecting PROMs for use within radiotherapy and PBT research.

## 5. Conclusions

This review highlights that a wide range of PROMs are used in radiotherapy and PBT research; however, most are not specific to radiotherapy-related side-effects and do not capture common treatment impacts, like radiation-induced skin reactions. Future research should focus on the development of radiotherapy-specific measures to use within photon vs. proton clinical trials or updating the historical questionnaires that are currently being utilised to encompass the other common impacts of radiotherapy on patients if they are being used in clinical trials in this population. This review is the first article to provide context for clinicians and researchers to aid them in choosing the most appropriate PROM for their radiotherapy study cohort, as well as identifying the clinical need for future PROM development.

## Data Availability

No new data were created or analyzed in this study. Data sharing is not applicable.

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
