# Peer review of "Do Patient-Reported Outcome Measures (PROMs) Used Within Radiotherapy Clinical Trials Reflect the Impact of Treatment?"

_cancers, 2024, doi:10.3390/cancers16223832_

Round 1

Reviewer 1 Report

Comments and Suggestions for Authors

The authors provided a paper about “Do Patient Reported Outcome Measures (PROMs) used within radiotherapy clinical trials reflect the impact of treatment?”.

The topic is extremely relevant because even though frequently discussed in the larger oncological field only a few PROMs have been specifically designed to account for QoL in the setting of radiation oncology.

I have a few suggestions to improve the scientific soundness of the article:

1)      In the materials and methods section it would important to address how the search was performed to identify the 51 PROMs

2)      Again in the materials and methods section it would important to add hoe the additional side effects were selected from the consent forms and guidelines published by the UK Royal College of Radiologists (how many researchers were involved? How were disputes resolved among researchers?)

3)      The conclusion section is missing provide your suggestions to improve this clinical scenario and to foster the development of questionnaires specifically designed for radiation oncology

Author Response

Thank you reviewer for taking the time to review this work and providing your constructive comments. These have been addressed within the article, with comments addressing each suggestion below. 

Reviewer 1: 

I have a few suggestions to improve the scientific soundness of the article: 

  1. In the materials and methods section it would important to address how the search was performed to identify the 51 PROMs 

Thank you for this comment, this has now been added into the methods section. 

2. Again in the materials and methods section it would important to add hoe the additional side effects were selected from the consent forms and guidelines published by the UK Royal College of Radiologists (how many researchers were involved? How were disputes resolved among researchers?) 

Thank you for this comment, this has now been added into the methods section. 

3. The conclusion section is missing provide your suggestions to improve this clinical scenario and to foster the development of questionnaires specifically designed for radiation oncology 

Apologies for this oversite, this has now been completed.  

Reviewer 2 Report

Comments and Suggestions for Authors

Great work and important topics. as very innovative

I think as the topic of this research is innovative, i would recommend to emphasize the innovation of this research in abstract, and discussion and conclusion

Missing conclusion: please write up the conclusion

What will be the future study suggestions based on the findings of this study.

Methods (This is the major concern)

Please clarify the inclusion/exclusion criteria for article selections

How to evaluate the quality of the article, 

Who joined the article review,

Author Response

Thank you reviewer for taking the time to review this work and providing your constructive comments. These have been addressed within the article, with comments addressing each suggestion below.

Reviewer 2:  

1. I think as the topic of this research is innovative, i would recommend to emphasize the innovation of this research in abstract, and discussion and conclusion 

Thank you for this comment. We have added that this is the first review of its kind and emphasized the importance of this work. 

2. Missing conclusion: please write up the conclusion. What will be the future study suggestions based on the findings of this study. 

Apologies for this oversite, this has now been completed.  

3. Methods (This is the major concern). Please clarify the inclusion/exclusion criteria for article selections. How to evaluate the quality of the article, Who joined the article review. 

Thank you for this comment – we have now addressed the inclusion/exclusion criteria for the article selection, as well as article reviewers. In regard to the quality assessment of the articles, the purpose of the search was to identify PROMS used in studies and not how they had been used. Therefore, critical appraisal of articles themselves was not appropriate. However, we did critically appraise each of the PROMS using the COSMIN content validity guidelines, which has been reported in detail elsewhere [Fairweather et al., 2023 -DOI: 10.1016/j.radonc.2023.110071]. We have added text into the methods to this effect. 

Reviewer 3 Report

Comments and Suggestions for Authors

The Breast Cancer Treatment Outcomes Scale describes

patient-reported changes in the treated breast and surrounding

area compared with the contralateral breast which serves as an internal control (Stanton AL, Krishnan L, Collins CA. Form or function? Part 1.

Subjective cosmetic and functional correlates of quality of life in

women treated with breast-conserving surgical procedures and

radiotherapy. Cancer 2001;91:2273e2281).

The GemelliART (Gemelli Advanced Radiation Therapy)

developed the VALuation Endorsed by Oncological

Patient (VALEOþ) to monitor radiotherapy-related

side effects and the patient’s emotional status day by day.

A mobile app was developed by the Knowledge-Based

Oncology laboratories of the Gemelli ART, and it can be

downloaded on the Apple Store or as an Android App on

Google Play . ([VALuation Endorsed by Oncological Patient) KBO.com laboratories].

Available at: https://itunes.apple.com/us/app/valeo/

id1161469620?l5it&ls51&mt58; VALEOþ)

The Objective Breast Cosmesis Scale (OBCS) was introduced in a recent study yielding results similar to the results from a multidisciplinary/multi-gender committee. OBCS used non-standardized photographs which are the most widely used in documenting breast cosmesis during follow-up(Breast cancer 2020; 27:179-185; doi: 10.1007/s12282-019-01006-w.

The authors write "In this article, we instead aim to investigate the appropriateness of PROMs and their 62 use within radiotherapy clinical trials, focusing on trial endpoints. This comparison aims 63 to identify which PROMs are appropriate for use within radiotherapy and PBT clinical 64 trials, aiding clinicians and researchers in choosing the most appropriate PROM for their 65 study". Where is the comparison?? It is not clear

Author Response

Thank you reviewer for taking time to review this work and giving your constructive comments. We have now addressed these within the article where possible, and have provided responces to the suggestions below.

1. The Breast Cancer Treatment Outcomes Scale describes patient-reported changes in the treated breast and surrounding area compared with the contralateral breast which serves as an internal control (Stanton AL, Krishnan L, Collins CA. Form or function? Part 1. Subjective cosmetic and functional correlates of quality of life in women treated with breast-conserving surgical procedures and radiotherapy. Cancer 2001;91:2273e2281). 

The GemelliART (Gemelli Advanced Radiation Therapy) developed the VALuation Endorsed by Oncological Patient (VALEOþ) to monitor radiotherapy-related side effects and the patient’s emotional status day by day. A mobile app was developed by the Knowledge-Based Oncology laboratories of the Gemelli ART, and it can bedownloaded on the Apple Store or as an Android App on Google Play . ([VALuation Endorsed by Oncological Patient) KBO.com laboratories]. Available at: https://itunes.apple.com/us/app/valeo/id1161469620?l5it&ls51&mt58; VALEOþ) 

The Objective Breast Cosmesis Scale (OBCS) was introduced in a recent study yielding results similar to the results from a multidisciplinary/multi-gender committee. OBCS used non-standardized photographs which are the most widely used in documenting breast cosmesis during follow-up.  (Breast cancer 2020; 27:179-185; doi: 10.1007/s12282-019-01006-w. 

Thank you for highlighting this work. Unfortunately, none of these articles are eligible to be included in this article within the current scope of this research. The first paper including the Breast Cancer Treatment Outcomes Scale is from 2001 – our inclusion criteria is from 2008 onwards (to reflect modern day RT/PBT techniques). We cannot identify any radiotherapy or PBT clinical trial or observational studies which include the VALEOb app and therefore this is not included. The OBSC appears to be a software system for comparing photographs, and is not a patient reported outcome measure.  

2. The authors write "In this article, we instead aim to investigate the appropriateness of PROMs and their 62 use within radiotherapy clinical trials, focusing on trial endpoints. This comparison aims 63 to identify which PROMs are appropriate for use within radiotherapy and PBT clinical 64 trials, aiding clinicians and researchers in choosing the most appropriate PROM for their 65 study". Where is the comparison?? It is not clear 

Thank you for this comment, this has now been clarified within the introduction.